# Near-Edge X-Ray Absorption Fine-Structure Spectra and Specific Dissociation of Phe-Gly and Gly-Phe

**DOI:** 10.3390/ijms26062515

**Published:** 2025-03-11

**Authors:** Tse-Fu Shen, Yu-Ju Chiang, Yi-Shiue Lin, Chen-Lin Liu, Yu-Chiao Wang, Kuan-Yi Chou, Cheng-Cheng Tsai, Wei-Ping Hu

**Affiliations:** 1Scientific Research Division, National Synchrotron Radiation Research Center, Hsinchu 300092, Taiwan; 2International PhD Program for Science, National Sun Yat-sen University, Kaohsiung 80424, Taiwan; 3Department of Chemistry and Biochemistry, National Chung Cheng University, Chia-Yi 62102, Taiwancoim1197@gmail.com (K.-Y.C.);

**Keywords:** density functional theory, destination orbitals, quantum chemical calculation of NEXFAS spectra, specific dissociation, soft X-ray, core excitation, peptide structure

## Abstract

The total-ion-yield (TIY) near-edge X-ray absorption fine-structure (NEXAFS) spectra of two dipeptides were measured and analyzed to identify the excitation sites of core electrons and the corresponding destination molecular orbitals. Peptide molecules were transferred to the gaseous phase using traditional heating and MALDI methods, ensuring that the resulting NEXAFS spectra and fragmentation products were consistent across both approaches. Mass spectra obtained at various excitation energies revealed the branching ratios of products at each energy level, offering insights into specific dissociation phenomena. Notably, variations in excitation energy demonstrated a selective dissociation process, with certain products forming more efficiently. This specificity appears closely linked to dissociations near the peptide bond, where the nodal planes of destination molecular orbitals are located. These findings were validated using both small peptide models and peptoid molecules, highlighting consistent patterns in the dissociation behavior.

## 1. Introduction

Core-level excitation, X-ray photoelectron spectroscopy (XPS), and near-edge X-ray absorption fine-structure (NEXAFS) spectroscopy are powerful tools for investigating the electronic states of matter, owing to their element-specific and environment-sensitive capabilities [1,2,3,4,5]. Eberhardt and his co-workers pioneered a new research area—specific dissociation following core excitation—by combining mass spectroscopy with NEXAFS spectroscopy [6]. Subsequent research has investigated specific dissociation phenomena occurring in both the gaseous phase and on surfaces [7,8,9]. The enhanced production of dissociation products has been attributed to local excitation effects or the anti-bonding nature of destination orbitals coupled with rapid dissociation [7,10]. Research studies have also explored the specific dissociation in biological systems, highlighting its relevance in biomolecular research [11,12,13,14,15,16]. Combining enhanced-product-formation data with absorption spectra can provide insights into biomolecular structures.

Our previous research measured the branching ratio of peptide bond cleavage in aliphatic molecules with peptide bonds as high as 71%, following specific excitation to an anti-bonding orbital centered on the amide group [14]. The resolved branching ratios for each dissociation channel at specific excitation energies indicated that the enhanced product formation resulted from the specific dissociation [17]. This phenomenon can be explained by the significant weakening of the C–N bond in the peptide group due to the electron population in π* anti-bonding orbitals after core excitation. The C=O bond in the carbonyl group (either peptide or acid) also exhibits anti-bonding features in the destination orbitals, contributing to dissociation.

Specific dissociation has also been observed in aromatic molecules with lower branching ratios or less precise results [18,19,20]. In these cases, the mixed character and delocalization of destination orbitals on phenyl and amide groups reduced the specificity of peptide bond cleavage [21]. As a result, the branching ratios for peptide bond cleavage were below 35%, suggesting that functional group types and the orbital spatial distribution strongly influence dissociation specificity. Specific dissociation involves multiple peptide bonds in some peptide model molecules [22]. Similarly, specific dissociation has been observed around the amide group of core-excited peptoid molecules, which have structural similarities to peptides [23].

In this study, we investigated two dipeptide molecules, Gly-Phe and Phe-Gly, using core electron excitation, NEXAFS spectroscopy, and mass spectrometry. The molecular structures, destination orbitals, and spectral assignments were predicted through quantum chemical calculations. While the peptide bond (C–N bond) was expected to be the primary site for bond cleavage, the secondary cleavage sites included the C–C bonds in the phenyl ring, provided that π* orbitals were populated following core excitation. However, the resonance stabilization of the phenyl ring makes C–C bond cleavage less probable. Thus, the phenyl C–C and C=O bonds are regarded as the secondary cleavage sites. These findings are consistent with previous NEXAFS studies on peptide models and peptoids, where peptide bond cleavage was frequently observed [14,17,18,19,20,21,22,23]. The product distributions and the dissociation channels, particularly those associated with peptide bond cleavages, were analyzed. We also examined and compared the dependence of peptide bond cleavage branching ratios on the compositions of the model molecules.

## 2. Results and Discussion

### 2.1. Experimental and Theoretical NEXAFS Spectra

Figure 1 shows the calculated lowest-energy structures of Gly-Phe and Phe-Gly. The mass spectra were obtained at each photon energy by scanning the soft X-ray photon energy. The intensity of ionic products at each photon energy was used to derive the absorption intensity, yielding total-ion-yield (TIY) NEXAFS spectra, like Figure 2. In the case of core-excited peptide molecules analyzed using MALDI, the matrix molecules were also excited by X-rays, producing numerous ionic products. The Appendix A compares the heating and MALDI methods. The mass spectra of matrix molecules alone and combined with peptide molecules were measured separately, like Figure 3b. Only ions with mass-to-charge ratios distinct from those of matrix molecules were included in the NEXAFS analysis to isolate the ionic products of core-excited peptides. These spectra are referred to as partial-ion-yield (PIY) NEXAFS spectra.

#### 2.1.1. Gly-Phe

The experimental TIY and calculated carbon K-edge spectra of Gly-Phe are illustrated in Figure 2. The energy shift applied to the calculated spectrum is 10.0 eV for all peaks. The key destination orbitals are illustrated in Figure 4, and the NEXAFS peak assignments are listed in Table 1. The complete calculated destination orbitals are presented in Appendix A.

The measured spectrum of Gly-Phe reveals four significant peaks below 291.6 eV in the carbon K-edge region. The theoretical calculations identify distinct electronic transitions and the contributing carbon atoms for each peak. Notably, all virtual orbitals involved in these transitions exhibit Rydberg contributions, while only a subset of virtual orbitals demonstrates significant π* character localized on the phenyl group or peptide bond.

The first peak (1π*+Rb) centered at 285.2 eV corresponds to the transition of carbon 6 (C6) (on the phenyl ring, Figure 2) to V(2). The virtual orbital V(2) shows mixed characteristics, with significant π* contributions localized on the peptide bond and phenyl ring accompanied by delocalized Rydberg components.

The second peak (2π*+Rb) at 287.4 eV appears as a lower-energy shoulder of the prominent 3π*+Rb peak and is dominated by the C10 → V(7) transition. The V(7) orbital exhibits subtle π* characteristics localized on the peptide bond with notable Rydberg contributions.

The third peak (3π*+Rb) at 288.4 eV corresponds to the C9-to-V(1) transition. The V(1) orbital contains a nodal plane on the peptide bond with a significant Rydberg character distributed across the acid and phenyl groups.

The fourth peak (4π*+Rb) at 290.8 eV is dominated by the C8-to-V(25) transition. The V(25) orbital exhibits a significant Rydberg character throughout the molecule with weak anti-bonding contributions on the phenyl ring.

#### 2.1.2. Phe-Gly

The experimental TIY and the calculated carbon K-edge spectra of Phe-Gly are illustrated in Figure 5. An energy shift of 10.1 eV was applied to the calculated spectrum for all peaks. The key destination orbitals are illustrated in Figure 6, and the NEXAFS peak assignments are listed in Table 2. The complete calculated destination orbitals are provided in Appendix A. The NEXAFS spectrum of Phe-Gly reveals two prominent peaks below 291.6 eV in the carbon K-edge region. Similarly to Gly-Phe, all virtual orbitals exhibit Rydberg contributions, but only specific orbitals demonstrate significant π* character localized on the phenyl ring, carbonyl group, or peptide bond.

The first peak centered at 285.1 eV is associated with C4 → V(3) and C6 → V(2) transitions. As seen in Figure 6, V(3) is primarily a π* orbital localized on the phenyl ring, and V(2) shows π* character on the peptide bonds and Rydberg contributions around the phenyl ring. Both C4 and C6 are on the phenyl ring.

The smaller shoulder peak at 287.3 eV is primarily attributed to the C9 (carbonyl carbon of the peptide bond) → V(6) transition. The V(6) orbital is predominantly Rydberg in nature.

The highest peak centered at 288.3 eV is dominated by the C11 (acid carbonyl) → V(1) transition. V1 contains small π* contributions on the acid carbonyl group, with delocalized Rydberg character and a nodal plane between the beta carbon and phenyl ring.

Below 291 eV, the experimental and TD-DFT calculation spectra show strong agreement in terms of excitation energies and relative peak intensities. Despite containing the same amino acids, the reversed sequence alters the electronic structure and core-excitation features. Gly-Phe exhibits a more sharply defined main aromatic π* resonance and pronounced secondary shoulders, including the broad peak positions at 290.8 eV, which are absent in Phe-Gly. By contrast, Phe-Gly shows slightly shifted peak positions, broader line shapes, and modified intensity ratios. These subtle but reproducible spectral variations highlight that reversing the amino acid sequence modifies the local electronic environment, affecting the observed NEXAFS signatures.

### 2.2. Dissociation Pathways and Specific Dissociations of Gly-Phe

The dissociation pathways of core-excited Gly-Phe were deduced from the mass spectrum shown in Figure 3a and predictions based on the two well-studied peptide models acetanilide and N-benzylacetamide, which include phenyl rings [21]. In these models, the deuterium and ^15^N substituted molecules were used to identify ionic product compositions, facilitating the determination of dissociation pathways. Similar methods were employed here to predict dissociation pathways for Gly-Phe. Ionic products with *m*/*z* < 40 u involve multiple dissociation pathways, as discussed in our previous studies [21,22,23]. The products with *m*/*z* ≈ 43 u, observed to have high abundances, were assigned to terminal COCH_3_^+^ fragments from peptide bond cleavage. Similarly, products with *m*/*z* = 44–45 u were assigned to terminal COOH^+^ fragments. Products with *m*/*z* = 42–43 u likely correspond to fragments containing the C=O group combined with either carbon or nitrogen atoms and a few hydrogen atoms.

The ionic products with *m*/*z* = 51 u, which exclude oxygen, are assigned to either C_3_NH^+^ (from peptide bond cleavage) or C_4_H_3_^+^ (from ring-opening reactions). Figure 3a shows that the ionic products with *m*/*z* ≈ 65 u resemble those from core-excited N-benzylacetamide. Therefore, we assigned the same dissociation channel to these products, originating from the phenyl ring but missing one carbon atom. The ionic products with *m*/*z* ≈ 77 u and 91 u are assigned to fragments corresponding to the phenyl ring and one carbon-added phenyl ring, respectively. The ionic products with *m*/*z* ≈ 103 u correspond to the phenyl ring combined with two additional carbon atoms, likely due to the breaking of two C–C bonds. Other products with high mass-to-charge ratios exhibit low intensities and are not discussed here.

In our previous studies, branching ratios of ionic products as a function of excitation energy have been used to characterize the specific dissociations [14,16,17,22,23]. Ionic products formed more efficiently at resonant excitation energies are defined as resulting from specific dissociations. The ionic products from core-excited Gly-Phe were also identified as the products of specific dissociations. The ionic product with *m*/*z* = 28 u and 30 u was enhanced at 285.2 eV and 288.4 eV, as shown in Figure 7a. Due to various possible dissociation mechanisms for *m*/*z* = 28 u, only the proposed dissociation pathway is shown in Figure 7b. Additionally, the product with *m*/*z* = 42 u was enhanced at ≈ 285.2 eV, while products with *m*/*z* = 65 u and 91 u showed enhancement at 285.2 eV and 288.4 eV, respectively. Figure 7b illustrates the possible dissociation pathways for these three ionic products resulting from specific dissociations.

The specific dissociation patterns observed for core-excited Gly-Phe using the MALDI method were consistent with those obtained using the heating method. However, the number of useful products was lower, and the signal-to-noise ratio was reduced using MALDI, especially due to the influence of matrix molecules. Therefore, the specific dissociation patterns observed with MALDI are not discussed here. It is worth noting that while both methods provide consistent results, the heating method yielded superior signal-to-noise ratios in this study, making it more effective for characterizing specific dissociations of Gly-Phe.

### 2.3. Dissociation Pathways and Specific Dissociations of Phe-Gly

The mass spectrum of ionic products from core-excited Phe-Gly is shown in Figure 8a. Due to structural similarities with Gly-Phe, the dissociation pathways of Phe-Gly follow a similar pattern to that of the peptide bond reversed. Therefore, only a brief explanation is provided here to avoid redundancy. The ionic products with *m*/*z* ≈ 28 u are primarily attributed to CO^+^ or CNH_3_^+^. Products with *m*/*z* ≈ 42 u are assigned to CONH^+^, formed by breaking the C–CO bond. The ionic products with *m*/*z* ≈ 51 u are either C_3_NH^+^ (resulting from peptide bond cleavage) or C4H3^+^ (produced from ring opening reactions). The ionic products with *m*/*z* ≈ 65 u, 77 u, 91 u, and 103 u originate from phenyl-ring cleavage, with 0–2 methyl groups added to the phenyl ring. The ionic products with *m*/*z* = 120 u are identified as phenyl-CH_2_CHNH_2_^+^, resulting from C–CO bond cleavage. Notably, this product is more abundant in the MALDI method compared to the heating method and Gly-Phe, as shown in Figure 8b. In the MALDI experiments, the background from matrix molecules was excluded.

Specific dissociations of Phe-Gly were observed at resonant excitation energies, as shown in Figure 9a. Ionic products with *m*/*z* = 29 and 30 u were enhanced at ~285.1 eV, and two possible dissociation pathways for these products are illustrated in Figure 9b. For the ionic product with *m*/*z* = 42 u, three possible dissociation pathways are depicted in Figure 9b by blue circles. Ionic products with *m*/*z* = 63 u, 65 u, and 91 u were enhanced at 285.1 and 288.3 eV, with the dissociation mechanisms linked to the phenyl ring.

Although the abundance of ionic products with *m*/*z* = 120 u is low, it provides evidence for specific dissociation at about 288.7 eV. This product is formed by peptide bond cleavage, and the positive charge is localized on the phenyl-ring side. The phenomena of the specific dissociation from Phe-Gly using the MALDI method are consistent with the results of the heating method. However, the available products are fewer, and the signal-to-noise (S/N) ratios are lower. Therefore, the specific dissociation patterns are not discussed in detail here.

### 2.4. Specific Dissociations of Peptides and Predictions

The dissociation mechanisms of the two core-excited dipeptides can be categorized into two groups: those related to the phenyl ring and those related to the peptide linkage. This study defines the peptide linkage as a C–CO–NH–C structure, with all three bonds contributing to the specific dissociations observed in the two core-excited dipeptides. These dissociations occur at two resonant excitation energies—285.2 eV and 288.4 eV for Gly-Phe and 285.1 eV and 288.3 eV for Phe-Gly, particularly at lower resonant excitation energies.

For Gly-Phe, the molecular orbital V(2) (Figure 4) is the primary destination orbital at 285.2 eV. For Phe-Gly, V(2) and V(3) (Figure 6) are the primary destination orbitals at 285.1 eV. Specific dissociations at the peptide bond have been prominently observed in core-excited peptide model molecules [14,21,22]. Small peptoid molecules show selective dissociation of the C–CO bond at the nitrogen and oxygen K-edges and the N–CO bond at the carbon K-edge [23]. The theoretical calculations provided accurate absorption energies and corresponding destination orbitals. These orbitals suggest that the most probable sites for specific dissociation are chemical bonds where nodal planes are located. Specific dissociation primarily occurs around the peptide linkages in peptide models and dipeptides. Therefore, based on the observed products of specific dissociation, calculated resonant excitation energies, and destination molecular orbitals, the location of the particular dissociation can be narrowed down to a few more possible chemical bonds. This phenomenon simplifies the structural recognition of peptides before dissociation.

## 3. Materials and Methods

### 3.1. Experimental Section

The details of the experimental methods have been discussed in our previous studies [18,20]. Therefore, brief experimental details are provided in this section. To measure ionic products, a reflectron time-of-flight mass spectrometer (OA-R-TOF MS) with orthogonal acceleration located at BL05B1 in the National Synchrotron Radiation Research Center was used. During the experiments, two vaporization methods were used to transfer the condensed dipeptide molecules to the gaseous phase.

In the first method, dipeptides were stored in a sample cell heated to temperatures of up to 500 K and positioned 2 mm below the synchrotron radiation path. The aperture for molecular diffusion measured 2 mm × 22 mm. During the experiments, Gly-Phe and Phe-Gly were heated to 144.9 °C and 101.4 °C, respectively. The temperature was monitored using a vacuum ultraviolet (VUV) lamp (E-Lux 126, Excitech, Schortens, Germany). Ionization of the vaporized Gly-Phe and Phe-Gly molecules ensured stable parent ion intensities for at least three hours, and the temperature of each compound was also optimized for the highest S/N ratio. A rectangular aperture (8.8 mm × 40 mm) separated the ionization and detection chambers. The ionization and detection chamber pressures during the experiments were maintained at 6.0 × 10^−7^ Torr and 1.4 × 10^−7^ Torr, respectively.

The second method utilized matrix-assisted laser desorption/ionization (MALDI) to vaporize biomolecules. Although MALDI is typically used to generate biomolecule ions, this study focused on neutral biomolecules after removing the generated ions. The matrix and dipeptide molecules were mixed and sprayed onto a circular stainless steel sample plate fixed on linear and circular stages, as shown in Figure 10. Unlike typical MALDI experiments, the ratio of dipeptide molecules to matrix compounds was approximately 1:3 to generate a sufficient quantity of neutral dipeptide molecules in the gaseous phase. After removing the generated ions from MALDI, neutral molecules were ionized with a VUV lamp. An example of mass spectra from this process is included in the Appendix A.

Soft X-rays from synchrotron radiation were directed into the ionization chamber using a pair of bendable Kirkpatrick–Baez refocusing mirrors coated with gold. Molecules ionized by X-ray photons dissociated into ionic fragments due to the considerable internal energy left after Auger decay. These ionic fragments were collimated using a set of ion lenses in the ionization chamber and directed into the reflectron time-of-flight mass spectrometer [20,24,25,26,27], with orthogonal acceleration in the detection chamber. When the ion acceleration region was filled with the collimated ions, a pulsed voltage was applied to drive them into mass analysis. The repetition rate of the pulsed voltage was in the range of 40–57 kHz. Ions were detected using a micro-channel plate (MCP) detector (95 mm × 42 mm). The output signal from the MCP was amplified with a preamplifier and recorded using a rapid time-of-flight multiscalar device (FAST ComTec, model no. P7888, Oberhaching, Germany) with a 2 ns bin width.

The synchrotron radiation beamline, delivering > 10^12^ photons/s with an energy range of 60–1400 eV, has been reported previously [28]. For this study, only a brief description for the synchrotron radiation beamline is provided in this section. The energy resolutions at the carbon K-edges were approximately 200 meV for both dipeptide molecules. Experiments at the nitrogen and oxygen K-edges were attempted, but poor signal-to-noise ratios and interference were observed from water vapor, which limited our results. Even after 10 h in a vacuum, water vapor signals persisted. Consequently, the ionic product intensities with *m*/*z* = 16–18 u were excluded as they were attributed to water rather than dipeptides. The beam spot size was 0.4 mm × 0.2 mm at the focal point, located approximately at the center of the diffusive molecular beam. Photon energy was calibrated using CO_2_ absorption spectra at 290.77 eV [29].

Phe-Gly (purity > 98%) was purchased from Sigma Aldrich, Gly-Phe (purity 99.79%) was purchased from Chem-Impex, and both were used without further purification. For the heating method, the sample cell was placed in a vacuum chamber for more than five hours before the experiments.

### 3.2. Theoretical Method

The electronic ground-state structures of the two dipeptides were calculated using the B3LYP [30] density functional with the 6-31+G(d,p) basis set. A conformational search was conducted to identify the lowest-energy structures, which were then used for the NEXAFS calculations. In previous studies [22], we demonstrated that Time-Dependent Density Functional Theory (TDDFT) calculations, with appropriate energy shifts, are practical tools for predicting and analyzing core-excitation spectra.

In this study, the NEXAFS spectra were calculated at the TD-B3LYP/6-31+G(d,p) level of theory. Core binding energies were estimated using the same approach as that proposed in our previous study [22]. Molecular structure calculations were performed using the Gaussian 16 program [31], and NEXAFS spectral predictions were performed using the Q-CHEM 4.3 program [32].

## 4. Conclusions

The dissociation phenomenon of two neutral dipeptide molecules (Gly-Phe and Phe-Gly) was investigated to understand how specific dissociation occurs in peptide molecules. The NEXAFS spectra measured using the TIY method and calculated at the TD-B3LYP/6-31+G(d,p) level of theory at the carbon K-edges were consistent and were used to explain the specific dissociation that occurs after resonant excitations. The spectra of the two molecules are similar but with subtle differences. Several peptide model molecules were analyzed to identify the patterns of specific dissociations in core-excited peptides. In this study, the peptide linkage, represented as a C–CO–NH–C structure, was examined, with all three bonds contributing to the specific dissociations directly observed in the two core-excited dipeptides. These specific dissociation patterns are similar to those observed in most peptide models, particularly peptoid molecules. Because the cleavage of different peptide bonds produces distinct ionic products, these products could, in principle, be used to reconstruct the compositions and structures of the parent peptide molecules. Thus, the current study paves the way for applying dissociation patterns to peptide molecules and for future investigations into the mechanisms of oligopeptide photodissociation.

## Figures and Tables

**Figure 1 ijms-26-02515-f001:**
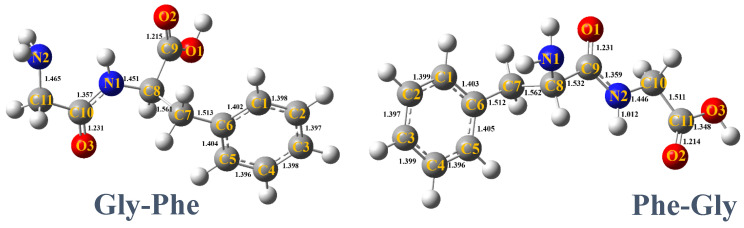
The lowest-energy structures of the Gly-Phe and Phe-Gly calculations using the B3LYP/6-31+G(d,p) method, including bond distances in angstroms.

**Figure 2 ijms-26-02515-f002:**
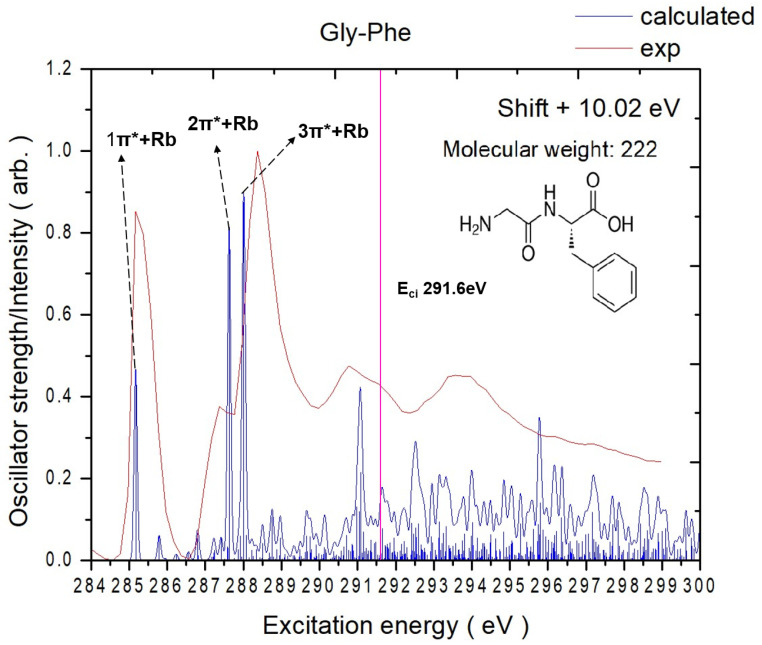
Experimentally measured (in total-ion-yield mode) and calculated NEXAFS spectra (after energy shift) of Gly-Phe at the carbon K-edges. The pink solid line represents the calculated core ionization potential, labeled as E_ci_.

**Figure 3 ijms-26-02515-f003:**
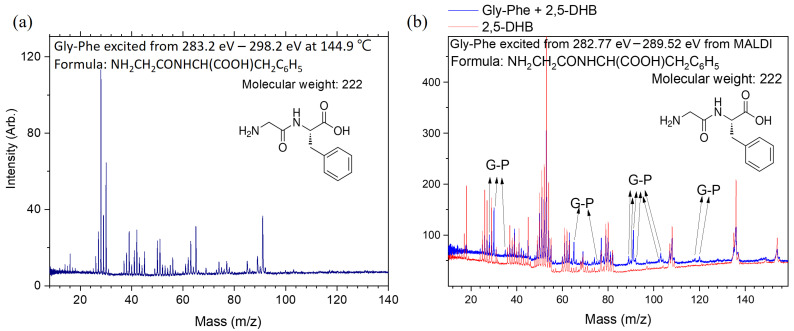
Mass spectra of Gly-Phe dissociation following core excitation and subsequent Auger decay at the carbon K-edges. (**a**) Heating the sample cell at 106 °C. (**b**) MALDI method.

**Figure 4 ijms-26-02515-f004:**
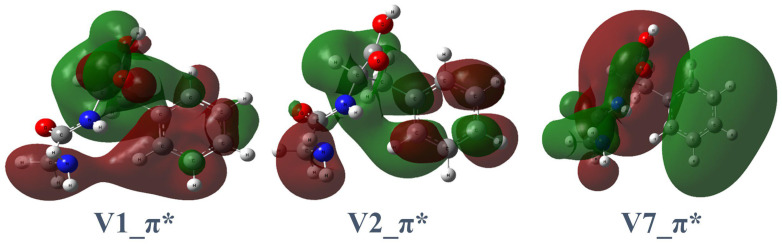
Three core-excitation destination orbitals of Gly-Phe involved in specific transitions. The two colors mean different signs (positive or negative) for the values of molecular orbitals. The intersections between two colors are nodal planes which have zero values. For destination orbitals, the nodal planes indicate the regions where chemical bonds are easier to break.

**Figure 5 ijms-26-02515-f005:**
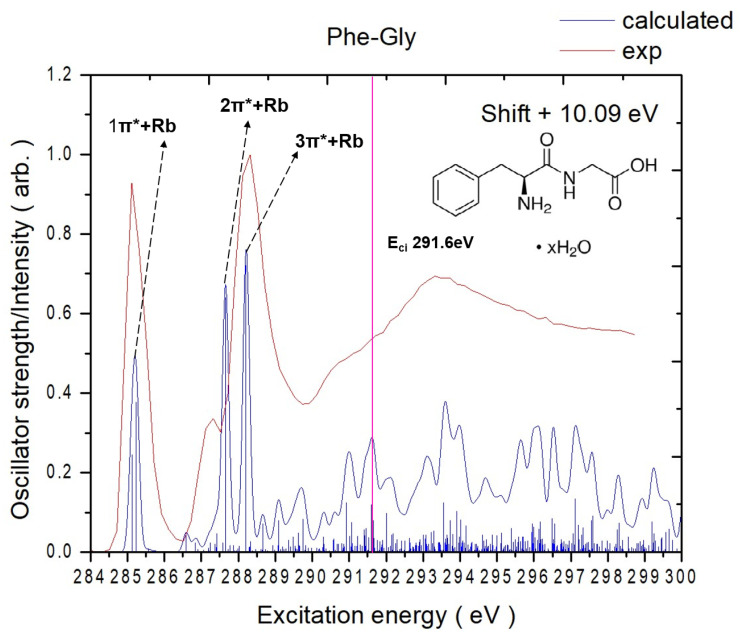
Experimentally measured (in total-ion-yield mode) and calculated NEXAFS spectra (after energy shift) of Phe-Gly at the carbon K-edges. The pink solid line represents the calculated core ionization potential, labeled as E_ci_.

**Figure 6 ijms-26-02515-f006:**
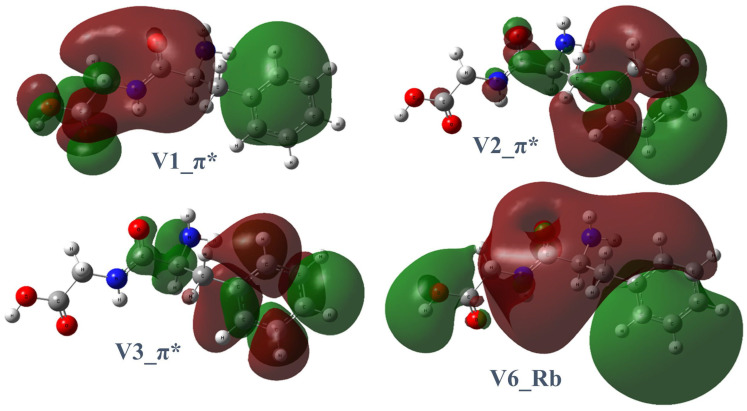
Four core-excitation destination orbitals of Phe-Gly involved in specific transitions.

**Figure 7 ijms-26-02515-f007:**
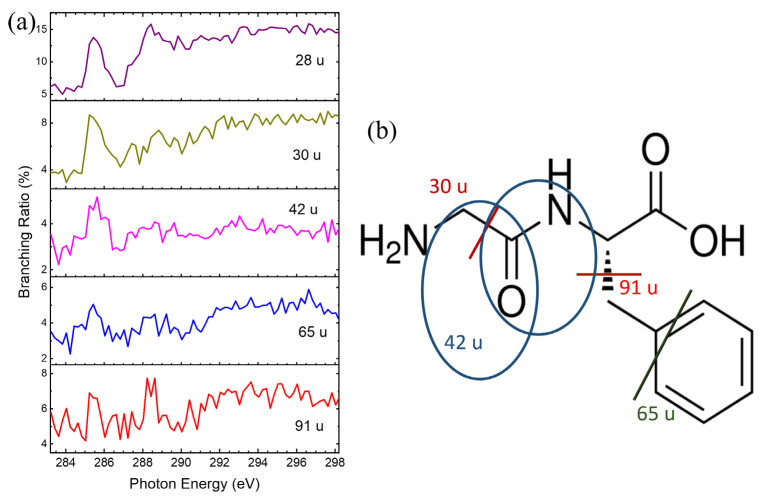
(**a**) Branching ratios of key ionic products formed from the core-excited Gly-Phe as a function of photon energy. (**b**) Proposed specific dissociation pathways of core-excited Gly-Phe following core excitation 1s → π* at the carbon K-edges. The different colored lines/circles and the corresponding colored m/z show the products from the dissociation pathways.

**Figure 8 ijms-26-02515-f008:**
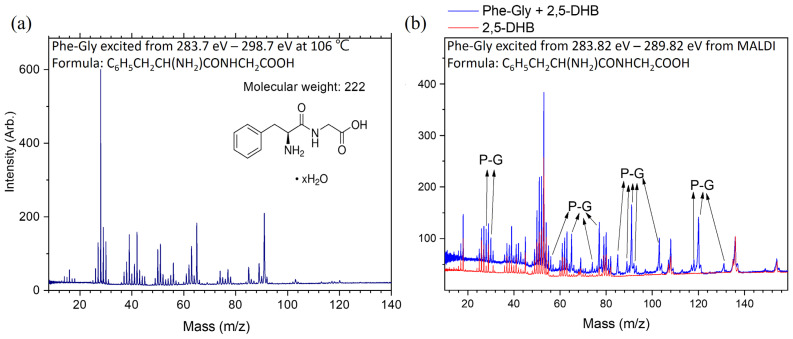
Mass spectra of Phe-Gly dissociation following core excitation and subsequent Auger decay at the carbon K-edges. (**a**) Heating the sample cell at 106 °C. (**b**) MALDI method.

**Figure 9 ijms-26-02515-f009:**
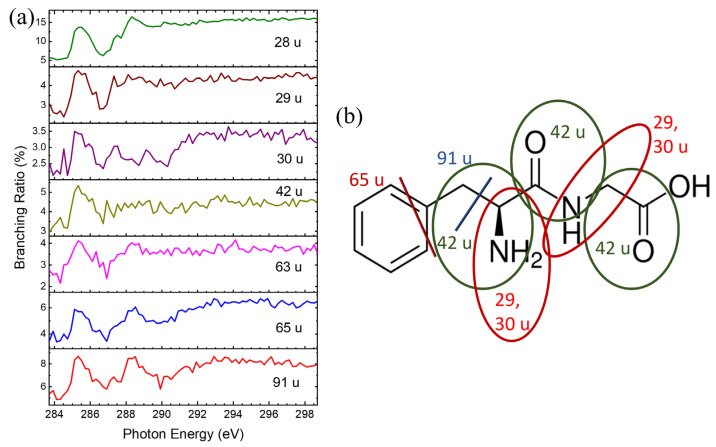
(**a**) Branching Ratios of essential ionic products formed from core-excited Phe-Gly as a function of photon energy. (**b**) Proposed dissociation pathway of core-excited Phe-Gly following core excitation 1s → π* at the carbon K-edges. The different colored lines/circles and the corresponding colored m/z show the products from the dissociation pathways.

**Figure 10 ijms-26-02515-f010:**
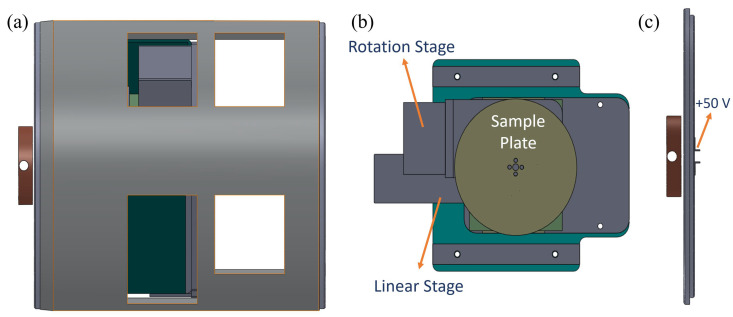
A schematic representation of the MALDI system setup. (**a**) The assembly of sample components in the MALDI system. (**b**) The assembled sample plate mounted on rotation and linear motion platforms for precise positioning. (**c**) Two small electrodes positioned to remove generated ions before the vaporized molecules enter the ionization chamber. A voltage of ~50 V was applied to the electrodes for ion removal.

**Table 1 ijms-26-02515-t001:** Assignments of the NEXAFS carbon K-edge spectra of Gly-Phe based on theoretical calculations.

Assignments of the NEXAFS K-Edge Spectrum of Gly-Phe				
Peak	Calculated Core-Excitation Energy (eV) ^*a*^	Corresponding Expt. Peak (eV) ^*b*^	Transition Orbitals	Normalized Oscillator Strength
Carbon K-edge				
1π*+Rb	285.2	285.2	C6 → π* (V2)	0.0352
	285.8		C7 → π* (V1)	0.0046
	286.8		C11→ π* (V3)	0.0046
	287.4		C7 → Rb (V8)	0.0042
2π*+Rb	287.6	287.4	C10 → π* (V7)	0.0577
	287.9		C8 → π* (V4)	0.0020
	287.9		C11 → Rb (V6)	0.0045
3π*+Rb	288.0	288.4	C9 → π* (V1)	0.0658
	288.1		C7 → Rb (V12)	0.0046
	288.3		C11 → π* (V10)	0.0020
	288.5		C11 → π* (V9)	0.0032
4π*+Rb	291.1	290.8	C8 → π* (V25)	0.0162

^*a*^ Calculated values + 10.0 eV. ^*b*^ Experimentally measured values.

**Table 2 ijms-26-02515-t002:** Assignments of the NEXAFS carbon K-edge spectra of Phe-Gly based on theoretical calculations.

Assignments of the0 NEXAFS K-Edge Spectrum of Phe-Gly				
Peak	Calculated Core-Excitation Energy (eV) ^*a*^	Corresponding Expt. Peak (eV) ^*b*^	Transition Orbitals	Normalized Oscillator Strength
Carbon K-edge				
1π*+Rb	285.1	285.1	C4 → π* (V3)	0.0228
	285.2		C6 → π* (V2)	0.0352
	286.6		C4 → Rb (V5)	0.0046
	287.2		C4 → Rb (V7)	0.0022
	287.4		C8 → π* (V4)	0.0044
2π*+Rb	287.7	287.3	C9 → Rb (V6)	0.0597
	287.9		C10 → π* (V4)	0.0017
	288.0		C8 → Rb (V8)	0.0014
3π*+Rb	288.2	288.3	C11 → π* (V1)	0.0673
	288.7		C10 → Rb (V8)	0.0066
	289.1		C10 → π* (V10)	0.0075

^*a*^ Calculated values + 10.1 eV. ^*b*^ Experimentally measured values.

## Data Availability

There are no data unavailable due to privacy or ethical restrictions.

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
