# Peer review of "Near-Edge X-Ray Absorption Fine-Structure Spectra and Specific Dissociation of Phe-Gly and Gly-Phe"

_ijms, 2025, doi:10.3390/ijms26062515_

Round 1

Reviewer 1 Report

Comments and Suggestions for Authors

This manuscript explores the total-ion-yield near-edge X-ray absorption fine structure spectra of two di-peptides and the following fragmentation of the studied species. It is a correctly carried out work and deserves to be published.

I have just one comment, which does not require to be answered but rather considered for the future work:

The authors have used rather old method for their calculations – B3LYP/6-31+G**. Nowadays, mush better functionals and bigger basis sets are available with realtively low computational cost, so those should be used instead.

Author Response

Comment 1: This manuscript explores the total-ion-yield near-edge X-ray absorption fine structure spectra of two di-peptides and the following fragmentation of the studied species. It is a correctly carried out work and deserves to be published.

Reply 1: We thank the recommendation from reviewer to publish this research.

Comment 2: I have just one comment, which does not require to be answered but rather considered for the future work: The authors have used rather old method for their calculations – B3LYP/6-31+G**. Nowadays, mush better functionals and bigger basis sets are available with relatively low computational cost, so those should be used instead.

Reply 2: We thank the reviewer for the reminder on the new functionals and basis sets. We will be more than happy to use and benchmark their performance in future studies.

Reviewer 2 Report

Comments and Suggestions for Authors

The analyzed article represents a continuation of the research carried out by the group of authors on the interactions of peptide structures. The study investigates the dipeptide structures Gly-Phe and Phe-Gly, focusing in particular on the dissociation of the peptide bond and those in its vicinity. In order to identify the transitions of the electrons involved and their belonging, spectral techniques (NEXAFS and MS) were combined with theoretical methods (TD-B3LYP).

The results obtained using these methods highlight the importance of the structural configurations of peptides by analyzing the dissociation energies of the peptide bond and adjacent bonds. The theoretical data accurately reproduce the experimental results, providing important information regarding the localization of the dissociation possibilities in the investigated systems. This type of investigations can be used in determining the structure and conformation of dipeptides.

I suggest to the authors to insert a distinct paragraph, Conclusions, at the end of the article or to modify the title of subparagraph 4.4 accordingly, to more clearly emphasize the main conclusions and the importance of the study conducted.

Author Response

Comment 1: I suggest to the authors to insert a distinct paragraph, Conclusions, at the end of the article or to modify the title of subparagraph 4.4 accordingly, to more clearly emphasize the main conclusions and the importance of the study conducted.

Reply 1: We thank the reviewer for this suggestion. We have added a paragraph as conclusions following the discussion part of this manuscript to emphasize the importance of this study.

Reviewer 3 Report

Comments and Suggestions for Authors

This manuscript investigates the dissociation of Phe-Gly and Gly-Phe molecules using XANES (NEXAFS) at the C K-edge. The study builds upon the methodologies of previous research [18, 19], maintaining a similar structure and incorporating feedback from earlier reviews.

I have a question regarding the interpretation of the NEXAFS spectra for Phe-Gly compared to Gly-Phe:

For Gly-Phe, the first peak is primarily attributed to C10, with Table 1 also indicating a minor contribution from C11. This explanation is reasonable, as it highlights a distinct geometric feature where the C-ring connects to the rest of the molecule.

However, for Phe-Gly, the first peak is attributed to both C11 and C10, with Table 2 showing a dominance of C10. C10 is in the same position as C10 in Gly-Phe, but C11 is in the C-ring part. Geometrically, C11 is nearly equivalent to C15, but there is no contribution from C15. At the same time, there is no contribution from C16 (C11 in Gly-Phe). Given the molecular geometry, a natural question arises: could there be an error?

This discrepancy should be corrected or explained.

Additionally, please improve the visibility of the atom numbers in Figure 1. It is really hard to find them without zooming in, especially in the printed version.

Author Response

Comment 1: This manuscript investigates the dissociation of Phe-Gly and Gly-Phe molecules using XANES (NEXAFS) at the C K-edge. The study builds upon the methodologies of previous research [18, 19], maintaining a similar structure and incorporating feedback from earlier reviews. I have a question regarding the interpretation of the NEXAFS spectra for Phe-Gly compared to Gly-Phe: For Gly-Phe, the first peak is primarily attributed to C10, with Table 1 also indicating a minor contribution from C11. This explanation is reasonable, as it highlights a distinct geometric feature where the C-ring connects to the rest of the molecule. However, for Phe-Gly, the first peak is attributed to both C11 and C10, with Table 2 showing a dominance of C10. C10 is in the same position as C10 in Gly-Phe, but C11 is in the C-ring part. Geometrically, C11 is nearly equivalent to C15, but there is no contribution from C15. At the same time, there is no contribution from C16 (C11 in Gly-Phe). Given the molecular geometry, a natural question arises: could there be an error? This discrepancy should be corrected or explained.

Reply 1: We sincerely thank the reviewer to point out the possible discrepancies on the peak assignment. After checking with the theoretical calculation, we did find an error in the atom labeling in Figure 2. The C11 position of Phe-Gly was labeled wrong inside the phenyl ring. To avoid the confusion between the carbon number and the orbital number of the origin of excitation, we have decided to rename the transition using the excited carbon number directly for clarity, such as C6 -> π* (V2). Thus, the dominant contribution to the first peak for GLY-Phe is now C6 -> V2 where C6 is the carbon in the phenyl ring that connects to the side chain of Phe. The revised atom numbering is shown in Figure 2. For Phe-Gly, the dominant contribution to the first peak is also C6 -> V2 (consistent with Gly-Phe) and with significant contribution from C4 -> V3 which is unique to Phe-Gly. Tables I and II and relevant text in the manuscript have been revised accordingly.

Comment 2: Additionally, please improve the visibility of the atom numbers in Figure 1. It is really hard to find them without zooming in, especially in the printed version.

Reply 2: We think the reviewer is suggesting the visibility of the atom numbers in Figure 2. So we have enlarged the atom numbers in Figure 2 as suggested by the reviewer.

Round 2

Reviewer 3 Report

Comments and Suggestions for Authors

The manuscript is improved and can be published.